# Preparation and Performance Evaluation of Amphiphilic Polymers for Enhanced Heavy Oil Recovery

**DOI:** 10.3390/polym15234606

**Published:** 2023-12-02

**Authors:** Dongtao Fei, Jixiang Guo, Ruiying Xiong, Xiaojun Zhang, Chuanhong Kang, Wyclif Kiyingi

**Affiliations:** 1Unconventional Petroleum Research Institute, China University of Petroleum, Beijing 102249, China; fdt305@163.com (D.F.); zxjhuagong@163.com (X.Z.); k15616309001@163.com (C.K.); kiyingiwyclif@outlook.com (W.K.); 2College of Petroleum Engineering, China University of Petroleum, Beijing 102249, China; ryxiong08@163.com

**Keywords:** amphiphilic polymer, hydrophobic association, self-emulsification, heavy oil viscosity reduction, enhanced oil recovery

## Abstract

The continuous growth in global energy and chemical raw material demand has drawn significant attention to the development of heavy oil resources. A primary challenge in heavy oil extraction lies in reducing crude oil viscosity. Alkali–surfactant–polymer (ASP) flooding technology has emerged as an effective method for enhancing heavy oil recovery. However, the chromatographic separation of chemical agents presents a formidable obstacle in heavy oil extraction. To address this challenge, we utilized a free radical polymerization method, employing acrylamide, 2-acrylamido-2-methylpropane sulfonic acid, lauryl acrylate, and benzyl acrylate as raw materials. This approach led to the synthesis of a multifunctional amphiphilic polymer known as PAALB, which we applied to the extraction of heavy oil. The structure of PAALB was meticulously characterized using techniques such as infrared spectroscopy and Nuclear Magnetic Resonance Spectroscopy. To assess the effectiveness of PAALB in reducing heavy oil viscosity and enhancing oil recovery, we conducted a series of tests, including contact angle measurements, interfacial tension assessments, self-emulsification experiments, critical association concentration tests, and sand-packed tube flooding experiments. The research findings indicate that PAALB can reduce oil–water displacement, reduce heavy oil viscosity, and improve swept volume upon injection into the formation. A solution of 5000 mg/L PAALB reduced the contact angle of water droplets on the core surface from 106.55° to 34.95°, shifting the core surface from oil-wet to water-wet, thereby enabling oil–water displacement. Moreover, A solution of 10,000 mg/L PAALB reduced the oil–water interfacial tension to 3.32 × 10^−4^ mN/m, reaching an ultra-low interfacial tension level, thereby inducing spontaneous emulsification of heavy oil within the formation. Under the condition of an oil–water ratio of 7:3, a solution of 10,000 mg/L PAALB can reduce the viscosity of heavy oil from 14,315 mPa·s to 201 mPa·s via the glass bottle inversion method, with a viscosity reduction rate of 98.60%. In sand-packed tube flooding experiments, under the injection volume of 1.5 PV, PAALB increased the recovery rate by 25.63% compared to traditional hydrolyzed polyacrylamide (HPAM) polymer. The insights derived from this research on amphiphilic polymers hold significant reference value for the development and optimization of chemical flooding strategies aimed at enhancing heavy oil recovery.

## 1. Introduction

Petroleum is a vital energy source and chemical feedstock, constituting a global reserve of approximately 9–11 trillion barrels, with heavy oil and bitumen resources accounting for over 70% of the total reserves. Given the escalating demand for energy and chemical feedstocks, the development of heavy oil resources has garnered significant attention [1,2,3]. Heavy oil extraction is challenging due to its high viscosity and limited mobility, making viscosity reduction within the reservoir crucial for successful heavy oil recovery [4,5].

Chemical flooding technology has been proven to be an effective method to enhance crude oil recovery. This approach involves the application of alkali, surfactants, polymers, and other chemicals to reduce heavy oil viscosity, increase flooding fluid viscosity, improve the oil–water mobility ratio and expand the sweep volume of oil displacement, ultimately enhancing heavy oil recovery [6,7,8,9,10]. However, this complex system often experiences chromatographic separation within the reservoir, hindering effective oil recovery. In recent years, scholars have begun investigating the application of amphiphilic polymers in heavy oil recovery. These polymers, featuring both hydrophilic and hydrophobic groups, facilitate efficient emulsification of heavy oil, reducing the viscosity of the oil phase while increasing that of the water phase. This, in turn, enhances the sweep efficiency of oil displacement without suffering from chromatographic separation. Thus, amphiphilic polymers hold substantial promise for heavy oil recovery [11,12,13,14,15,16].

Li [16] synthesized a series of water-soluble long branched-chain amphiphilic copolymers, AAGASs, and applied in the enhanced oil recovery of heavy oil. Hydrophobically associating water-soluble copolymers (AAGs) with an epoxy group were first synthesized via the free-radical copolymerization of acrylamide and sodium 2-acrylamido-2-methylpropanesulfonic acid with glycidyl methacrylate. An amino-terminated amphiphilic copolymer (AS-N) was also prepared with acrylamide and sodium 4-styrenesulfonate via chain transfer polymerization. AAGASs were then obtained via the chain-extending reaction between the epoxy groups of the AAG and the amino group of the AS-N. The results revealed that AAGAS exhibited unique association properties in solution, displaying excellent surface and interfacial activity. Consequently, AAGAS solutions were capable of both thickening water and transforming high-viscosity heavy oil into low-viscosity oil-in-water emulsions. Notably, they achieved an impressive 96.8% reduction in heavy oil viscosity at a concentration of 1000 mg/L AAGAS-3.

Qin [17] prepared a dendrimer amphiphilic polymer (A-D-HPAM) via the aqueous solution polymerization of acrylic acid, acrylamide, 2-acrylamido-2-methylpropanesulfonic acid, hydrophobic monomer of cetyl dimethyl allyl ammonium chloride, and skeleton monomer. It exhibited experimentally proven superior polymer performance. Compared to hydrolyzed polyacrylamide and hydrolyzed polyacrylamide–sodium dodecyl benzene sulfonate solutions, A-D-PAM notably increased oil recovery by 15.1%. Chen [18] synthesized the amphiphilic polymer PTVR using acrylamide, sodium p-styrenesulfonate, lauryl methacrylate, and methyl-2-urea-4[1H]-pyrimidinone. The results highlighted the exceptional performance of PTVR, which was attributed to the synergistic effect of these monomers. Methyl-2-urea-4[1H]-pyrimidinone is a quadruple hydrogen bond monomer, which effectively disrupts the hydrogen bond interactions within asphaltene or resin aggregates. On the other hand, the long alkyl chain lauryl methacrylate improved the polymer’s affinity with heavy oil, facilitating the emulsification of heavy oil.

Liu [19] synthesized an amphiphilic polymer, AD-1, via free radical polymerization using acrylamide, acrylamide, 2-acrylamido-2-methyl-1-propane sulfonic acid, and self-tailored monomers with long hydrophobic chains and benzene rings, and conducted an investigation into amphiphilic polymer AD-1’s role in enhancing heavy oil recovery via hot water flooding. The findings demonstrated that AD-1 had the capacity to form stable aggregates and a three-dimensional network in solution, exhibiting excellent interfacial performance. Even at low concentrations, AD-1 could create dynamic water-in-oil emulsions, facilitating both emulsification and rapid demulsification. Yang [20] synthesized a betaine-type binary amphiphilic polymer, PAMA, from acrylamide and association monomer made in a laboratory. The study elucidated the mechanism by which varying hydrophobic group content influenced PAMA’s performance. Experimental findings demonstrated that higher hydrophobic group content improved PAMA’s ability to reduce surface tension and foster associations among hydrophobic groups. Increased hydrophobic group content enhanced the polymer’s viscosity, thermal stability, viscoelastic properties, and salt-thickening behavior.

Tian [21] conducted a comparative analysis of amphiphilic polymers featuring mono-branched and multi-branched hydrophobic groups. The research showed that multi-branched amphiphilic polymers had a dual effect, increasing both the viscosity and interfacial viscoelasticity of the dispersion medium, resulting in more stable emulsions. Zhu [22] examined amphiphilic polymers with both single-tail and double-tail hydrophobic groups to understand their impact on solution properties. The findings indicated that polymers with double-tail hydrophobic groups exhibited a lower critical association concentration (CAC). This led to stronger hydrophobic associations and, consequently, a higher apparent viscosity contribution rate when polymer concentrations exceeded the CAC. Moreover, amphiphilic polymers with double-tail hydrophobic groups displayed improved temperature resistance, salt resistance, and mechanical shear resistance at equivalent polymer solution concentrations. Zhi [23] employed amphiphilic polymer as a heavy oil activator and investigated the changes in residual oil distribution following activator displacement, along with its performance in heavy oil activation. The results demonstrated that the activator could reduce oil–water interfacial tension, break down and disperse recombination fractions in heavy oil, and decrease heavy oil viscosity. Post-activation, the utilization of residual oil in medium and small pores significantly improved, particularly in residual oil with film-like and cluster-like structures. This process slowed the decline rate of light/heavy fractions in heavy oil. The heavy oil activator effectively enhanced the sweep volume and displacement efficiency of heavy oil, thereby playing a crucial role in improving recovery rates in heavy oil reservoirs. Ma [24] synthesized a novel low-molecular-weight amphiphilic viscosity reducer. Employing Materials Studio 5.0 software, Ma explored the synthesis feasibility and viscosity reduction mechanism of this heavy oil viscosity reducer through a molecular dynamics perspective. The molecular dynamics simulations revealed that the addition of the viscosity reducer altered the potential energy, non-potential energy, density, and hydrogen bond distribution within the heavy oil. Specifically, the benzene ring in butadiene benzene became embedded in the asphaltene’s interlayer structure, effectively weakening the heavy oil’s network configuration. The results of numerous studies have shown that amphiphilic polymers prepared by introducing hydrophobic monomers into acrylamide-based copolymers have good effects on improving heavy oil recovery.

In this study, based on the cost-effectiveness of materials and the difficulty of synthesis, we selected acrylamide and 2-acrylamido-2-methylpropane sulfonic acid commonly used in oil fields as hydrophilic monomers and selected long-chain lauryl acrylate and benzyl acrylate with benzene ring as hydrophobic monomers to prepare amphiphilic polymers PAALB through free radical polymerization. The addition of acrylamide and 2-acrylamido-2-methylpropanesulfonic acid effectively improved the viscosity, temperature resistance and salt resistance of the polymer solution. The addition of monomers with long chains provided the polymers with interfacial activity. The addition of monomers containing benzene rings facilitates the dispersion of heavy oil components in the polymer, thus enhancing the interaction between the polymer and the heavy oil. The amphiphilic polymer has excellent interfacial activity, emulsifying ability and viscoelasticity, which can both reduce the viscosity of the oil phase and increase the viscosity of the aqueous phase. This enhancement changes the water-oil mobility ratio, increases sweep volumes, and contributes to the overall enhancement of heavy oil recovery. In addition, an important capability of our synthesized amphiphilic polymers is the ability to achieve self-emulsification of heavy oil. However, in the reports on spontaneous emulsification of crude oil, the viscosity of most crude oil is lower than 2000 mPa·s, and the spontaneous emulsification of heavy oil above 10,000 mPa·s is not reported [25,26,27,28,29]. Therefore, this amphiphilic polymer has better applicability to the self-emulsification and viscosity reduction of high-viscosity heavy oils.

## 2. Materials and Methods

### 2.1. Materials

The reagents used include acrylamide (AM, 99% pure, CAS No. 79-06-1), 2-acrylamido-2-methylpropansufonic acid (AMPS, purity 98%, CAS No. 15214-89-8), lauryl acrylate (LA, purity 90%, CAS No. 2156-97-0), benzyl acrylate (BA, purity 97%, CAS No. 2495-35-4), 2,2′-azobis (2-methylpropionamidine) dihydrochloride (AIBA, purity 97%, CAS No. 2997-92-4), sodium hydroxide (NaOH, purity 98%, CAS No. 1310-73-2), iron sulfate heptahydrate (FeSO_4_·7H_2_O, purity 99%,CAS No. 7782-63-0), hydrogen peroxide (H_2_O_2_, 30% aqueous solution, CAS No. 7722-84-1), edetate disodium (EDTA, purity 98%, CAS No. 6381-92-6), ethanol (purity 99.7%, CAS No. 64-17-5), and sodium dodecyl sulfate (SDS, purity 98%, CAS No. 151-21-3). These were sourced from Aladdin company (Shanghai, China).

Partially hydrolyzed polyacrylamide (HPAM) with a relative molecular weight of 1.2 × 10^7^ and hydrolysis degree of 20% was obtained from Zhangjiakou Shengda Polymer Co., Ltd. (Zhangjiakou, China). The heavy oil used in the experiments was sourced from Chunfeng Oilfield, and its physical properties are presented in Table 1. The mineralization composition of the brine water is shown in Table 2.

### 2.2. Synthesis of PAALB

The synthesis process of the PAALB polymer is illustrated in Figure 1. PAALB, a quaternary copolymer, was synthesized through free radical copolymerization. The process involved using AM, AMPS, LA, and BA as the raw materials. The monomers were measured and placed in a 250 mL three-necked flask equipped with a mechanical overhead stirrer and subjected to N_2_ purge gas. The molar ratio of AM:AMPS:LA:BA was 6:2:1:1. Deionized water was added to the flask to maintain the total monomer concentration in the solution at 30%. SDS (at 0.3% of the total monomer) was introduced to facilitate the dissolution of the hydrophobic monomer, and the solution’s pH was adjusted to 6 using an aqueous NaOH solution. Following a 30-min N_2_ purge, the solution was cooled to a temperature within 0–10 °C. Subsequently, 50 mg of AIBA, along with 300 ppm of EDTA, 500 ppm of FeSO_4_, and 400 ppm of H_2_O_2_, was added as an initiator, and the mixture was stirred for 30 min. This process resulted in a milky white viscous liquid. The temperature was then increased to 60 °C and maintained for 4 h, yielding the final product, PAALB. The product was washed with ethanol to remove impurities. It was subsequently dried in a vacuum oven at 60 °C for 24 h to obtain a solid product, which was then ground into a powder and stored [18]. The final polymer yield was 88.67%.

### 2.3. Polymer Characterization

The polymer powder was blended with KBr and compressed into a thin film. Subsequently, the Fourier transform infrared spectrum (FT-IR) was obtained using the Thermo Fischer Scientific Nicolet IS5 spectrometer (Thermo Fisher Scientific, Waltham, MA, USA).

A polymer solution with a concentration of 20 mg/mL was prepared by dissolving the polymer in D_2_O. The solution was added to the nuclear magnetic tube for scanning. The ^1^H nuclear magnetic resonance (NMR) spectroscopic analysis used the Bruker AVANCE III 600 NMR spectrometer (Bruker Corporation, New York, NY, USA).

C, H, O, N, and S quantities were determined using the Vario EL cube and Vario MACRO cube elemental analyzer (Elementar Germany, Frankfurt, Germany).

Gel permeation chromatography (GPC) was performed using an Agilent 1260 Infinity II LC liquid chromatograph equipped with an Agilent RID G1362A detector (Agilent Technologies Inc, Santa Clara, CA, USA) on a Waters Ultrahydrogel 300 × 7.8 mm 500-250-120A column. In this study, 0.1 mol/L NaNO_3_ aqueous solution was used as the mobile phase at a flow rate of 1 mL/min. The testing temperature was 40 °C, and the polymer sample concentration was 5000 mg/L.

### 2.4. Apparent Viscosity Measurement

The viscosity of the polymer solution to crude oil was determined using a HAAKE MARS III rheometer (Thermo Fisher Scientific, Waltham, MA, USA). The CC24Ti tumbler measuring system was selected for viscosity determination in a CR test mode during viscosity tests. The sample was sheared at 7.34 s^−1^ for 300 s at 50 °C, and the measurements were taken every 30 s. The viscosity–temperature curve was determined in a CR test mode at a shear rate of 7.34 s^−1^, with recordings made from 20 to 100 °C at a step of 3 °C per 60 s for a total test time of 1600 s.

### 2.5. Critical Association Concentration Measurement

This research employed pyrene as a fluorescence spectroscopy probe to investigate the hydrophobic association behavior of amphiphilic polymers. In the experiment, a pyrene ethanol solution of 2.5 × 10^−3^ mol/L was added to various polymer solution concentrations to achieve a final pyrene concentration of 5 × 10^−5^ mol/L (a concentration that had minimal impact on solution properties). The solutions were sealed, stirred, and left for 24 h. The fluorescence spectrum of pyrene was recorded using a Hitachi F4600 fluorescence spectrophotometer (HITACHI, Tokyo, Japan), with an excitation wavelength of 335 nm and an emission spectrum recording range of 350–450 nm. The I_1_/I_3_ ratio reflected the change in polarity within the microenvironment surrounding pyrene, enabling the determination of hydrophobic association strength in the amphiphilic polymer solutions.

### 2.6. Polymer Self-Emulsification Ability Evaluation

For emulsification analysis, a 30 mL glass bottle was used, combining 4.5 mL of 5000 mg/L polymer solution and 10.5 mL of heavy oil, resulting in an oil–water ratio of 7:3. The bottle was inverted 50 times, and the color of the solution was observed. A uniform black solution after inversion indicated successful emulsification of heavy oil and the polymer [25,26,27].

### 2.7. Viscosity Reduction Performance Measurement

Polymer solutions of varying concentrations were prepared using brine water. Under conditions of 50 °C and a 7:3 oil–water ratio, the glass bottle was inverted 50 times, and the viscosity of the emulsion post-inversion was measured to evaluate the polymer’s heavy oil viscosity reduction performance. The viscosity reduction rate was calculated using the following formula:(1)$$\Phi =\frac{\mu_{0}-\mu }{\mu_{0}}\times 100\%$$
where *Φ* is the viscosity reduction rate, *μ*_0_ is the initial viscosity of heavy oil, and *μ* is the viscosity of oil–water emulsion after adding polymer.

### 2.8. Emulsion Structure Testing

For the examination of the emulsion formed by heavy oil and the polymer solution after emulsification, a small amount was drawn using a pipette and placed onto a microscope slide. A cover slip was carefully placed over the droplet using tweezers. The sample was then positioned on the stage of a Zeiss Primo Star microscope for emulsion image observation with a magnification of 100×.

### 2.9. Interfacial Tension Measurement

Polymer solutions with various concentrations were prepared using brine water. The interfacial tension (IFT) between the polymer solution and heavy oil was measured using the SVT20N video spinning drop tensiometer. The test was conducted at a temperature of 50 °C and a rotation speed of 6000 r/min.

### 2.10. Contact Angle Measurement

Core slices with neutral wetting surfaces were soaked in a 5000 mg/L polymer solution for 24 h. Then, the slices were dried in a hot air sterilizing drying oven at 80 °C. The contact angle of water droplets on the core surface was measured using an SDC200 contact angle meter (Ningbo Precious Instruments Technology Co., Ltd., Ningbo, China) to determine any changes in wettability on the core surface.

### 2.11. Polymer Flooding Performance

#### 2.11.1. Sand-Packed Tube Flooding Experiments

A sand-packed tube flooding device was utilized to evaluate the oil displacement effects of HPAM and PAALB (Figure 2). The parameters of the sand-packed tube are shown in Table 3. The sand-packed tube was filled with quartz sand, saturated with water to measure the porosity, and injected with water to measure the permeability. After that, the sand-packed tube was saturated with oil and allowed to age for 24 h. Water was then injected to displace the oil until the water content exceeded 98%. Subsequently, polymer injection was performed to displace the oil until the water content exceeded 98%. The following step was subsequent water flooding. The experimental conditions were as follows: temperature of 50 °C, HPAM concentration of 15,000 mg/L (150 mPa·s), PAALB concentration of 10,000 mg/L (145 mPa·s), and injection speed of 0.2 mL/min.

#### 2.11.2. Microscopic Model Flooding Experiments

The microscopic oil displacement effects of HPAM and PAALB were evaluated using a microscopic visualization model etched on glass (Figure 3). The experimental procedure included injecting silicone oil and aging it for 12 h to treat the pore surface of the model as an oil-wet surface. Subsequently, the micro model was saturated with oil and allowed to age for 24 h. Water was then injected to displace the oil until no oil was produced at the outlet. At this point, the polymer was injected until no oil was produced at the outlet, allowing observation of the oil displacement by the polymer. The experimental conditions were as follows: temperature of 50 °C, HPAM concentration of 15,000 mg/L (150 mPa·s), PAALB concentration of 10,000 mg/L (145 mPa·s), and injection speed of 0.12 mL/h.

## 3. Results and Discussion

### 3.1. Characterization of PAALB

#### 3.1.1. FT-IR and ^1^H-NMR

Figure 4a shows the infrared spectra of the monomer and polymer. The spectrum of acrylamide has distinct stretching vibrational bands of amino (–NH_2_) at 3170 cm^−1^ and 3340 cm^−1^. This peak slows down and shifts in the polymer spectrum, appearing at 3215 cm^−1^ and 3331 cm^−1^. The peaks at 1075 cm^−1^ and 1232 cm^−1^ in the infrared spectra of 2-acrylamido-2-methylpropansufonic acid are attributed to the symmetric and asymmetric vibrational absorption peaks of the sulfonic acid group (–SO_3_^−^); this absorption peak in the polymer spectrum is displaced and appears at 1037 cm^−1^ and 1157 cm^−1^. The peaks at 2853 cm^−1^ and 2922 cm^−1^ in the spectrum of lauryl acrylate are the stretching vibrational bands of methylene (–CH_2_) and methyl (–CH_3_) groups, and this peak can also be observed in the polymer spectrum. The peaks at 3067 cm^−1^ and 3081 cm^−1^ in the spectrum of benzyl acrylate are the characteristic group frequencies of the hydrocarbon bond (C–H) stretching vibration of the benzene ring, and the peaks at 696 cm^−1^ and 737 cm^−1^ are the characteristic absorption peaks of the monosubstituted benzene ring. In addition, the absorption peaks of the carbonyl group (C=O) were observed at 1773 cm^−1^ and 1671 cm^−1^. Conclusively, the polymers’ spectra contain each reacting monomer’s characteristic peaks, proving that we successfully prepared the copolymer PAALB via monomer copolymerization [18,30,31].

To further confirm the molecular structure of PAALB, proton nuclear magnetic resonance spectroscopy was conducted, as displayed in Figure 4b. The observed peaks and their respective assignments are as follows: the peak at 1.46 ppm corresponds to the H of the methylene (–CH_2_) on the long chain of lauryl acrylate. The peak at 1.49 ppm corresponds to the H of the methyl (–CH_3_) on AMPS. The peak at 1.73 ppm corresponds to the H of the secondary methyl on the polymer main chain. The peak at 2.19 ppm corresponds to the H of the methine on the polymer main chain. The peak at 3.41 ppm corresponds to the H of the methylene connected to S on AMPS. The peak near 5.67 ppm corresponds to the H of the methine connected to the benzene ring on benzyl acrylate. The peak near 6.11 ppm corresponds to the H of the amino group (–NH_2_) on acrylamide. The peak at 7.45 ppm corresponds to the H of the benzene ring on benzyl acrylate [32,33]. The results obtained from IR and ^1^H-NMR confirm the successful synthesis of the intended product.

#### 3.1.2. Elemental Analysis

The results from the elemental analysis of the polymer are shown in Table 4. According to theoretical calculation, when the molar ratio of AM:AMPS:LA:BA is 6:2:1:1, the mass proportion of each element in the polymer is C (55.07%), H (7.57%), O (23.19%), N (9.02%) and S (5.15%), respectively. Although there is some deviation between the test results of element analysis and the theoretical calculation results, the overall results are similar, and the deviation is very slight. The test results show that the monomer composition of the polymer is consistent with the feed ratio.

#### 3.1.3. Relative Molecular Mass of PAALB and Its Distribution

Table 5 shows the relative molecular mass distribution of polymer PAALB. As can be seen from Table 5, the number average relative molecular mass (*M*_n_) of the polymer is 307,883, the weight average relative molecular mass (*M*_w_) is 679,306, the Z average relative molecular mass (*M*_z_) is 1,048,818, the Z + 1 average relative molecular mass (*M*_z+1_) is 1,333,611, and the viscosity average relative molecular mass (*M*_v_) is 623,887. The polymer’s relative molecular mass polydispersity index (PDI) was 2.21, and the test results proved the successful preparation of the polymer.

### 3.2. Critical Association Concentration of Polymer

Figure 5 is a display of viscosity change and critical association concentration (CAC) of the polymer solution in deionized and brine water. Specifically, Figure 5a,b display the fluorescence intensity of pyrene in PAALB solutions of deionized and brine water, respectively. The emission spectrum of pyrene, excited at 335 nm, displayed five emission peaks at 373 nm, 379 nm, 384 nm, 390 nm, and 410 nm. The intensity ratio of the first vibrational peak (I_1_) to the third vibrational peak (I_3_) served as an indicator of the polarity of the surrounding environment for pyrene molecules. A smaller I_1_/I_3_ ratio suggested a higher degree of hydrophobic association. Thus, the I_1_/I_3_ ratio was utilized as a measure for the degree of hydrophobic association for the amphiphilic polymers. When the I_1_/I_3_ ratio started to decrease, it indicated the occurrence of intermolecular hydrophobic association within the polymer, and this concentration was considered the CAC for the amphiphilic polymer. Figure 5c shows the change in the I_1_/I_3_ ratio of the pyrene vibrational peak with polymer solution concentration, indicating the CAC of PAALB to be 3500 mg/L in deionized water solution and 5000 mg/L in brine water. Figure 5d exhibits the viscosity change of the polymer in deionized water and brine water with increasing concentration. The viscosity initially increases slowly with concentration and then rapidly rises exponentially. This behavior is attributed to the hydrophobic association between polymer molecules, which leads to a sudden increase in viscosity when the concentration exceeds the CAC. Based on curve fitting and viscosity trends, the predicted CAC concentration of PAALB in distilled water is 3400 mg/L, while in brine water, it is 4300 mg/L. These predictions closely align with the experimental results presented in Figure 5c, validating the approximate prediction of CAC using the viscosity increase method [34,35].

### 3.3. Polymer Self-Emulsification Performance and Mechanism

#### 3.3.1. Polymer Self-Emulsification Performance

The progression of oil–water states with increasing numbers of glass bottle inversions can be observed in Figure 6. Initially, the polymer solution and heavy oil exhibited a clear separation. After a single inversion, oil droplets appeared in the polymer solution layer. The number of the droplets in the polymer layer increased as the inversions continued, and the color of the oil–water system darkened. A uniform oil-in-water (O/W) emulsion state was observed, where the oil phase was evenly distributed as droplets within the water phase. The results of the glass bottle inversion experiment demonstrate the ability of the PAALB polymer solution to emulsify heavy oil and form an O/W emulsion under weak shear forces generated via oscillation [25,26].

#### 3.3.2. Self-Emulsification Mechanism

The mechanisms underlying self-emulsification involve interfacial turbulence caused by Marangoni convection, the generation of negative interfacial tension values, and the diffusion and retention of chemical instability [36,37]. This paper considers the change in interfacial tension (IFT) during the dynamic self-emulsification process as the key factor influencing oil–water emulsification. Figure 7a depicts a container filled with deionized water, where water droplets in the liquid phase and at the gas-liquid interface experience different resultant forces. While water droplets in the liquid phase maintain force balance, those at the interface are subjected to a resultant force that directs them inward, causing the droplets to tend to escape the interface and enter the bulk phase. Similarly, individual droplets extracted from the liquid phase exhibit different forces acting on the surface and interior molecules. The resultant force on the surface molecules directs inward, maintaining the droplet’s spherical shape. Cutting a large droplet in half yields two smaller droplets, as the interfacial tension along the cutting line contracts the surface of the cut droplet, sustaining its spherical shape. This natural phenomenon is the result of interfacial tensions. Consequently, the interfacial tension at the oil–water interface promotes the contraction of water and oil droplets. High interfacial tension increases the tendency for droplet contraction. Introducing surfactants reduces the oil–water interfacial tension, diminishing the contraction tendency. As illustrated in Figure 7b, it can be noticed that as the interfacial tension is significantly increased, the oil and water remain immiscible; thus, an enormous amount of external energy is needed to overcome this interfacial tension for the mixing to occur. When the interfacial tension is significantly reduced, minimal shearing or disturbances to the system can overcome this interfacial tension and promote oil–water mixing. When the interfacial tension becomes negative, the force driving the contraction of the oil–water interface transforms into a force facilitating the diffusion of the two phases, thus their spontaneous mixing. Figure 7c depicts the process of spontaneous emulsification of the heavy oil and polymer solution when the glass bottle is inverted. This phenomenon occurs due to the polymer solution’s ability to reduce the interfacial tension between oil and water [38].

As shown in Figure 7d, oil–water interfacial tension decreases significantly with increasing polymer solution concentration. The oil–water interfacial tension without polymer addition was 11.43 mN/m. Upon reaching a PAALB concentration of 3000 mg/L, the oil–water interfacial tension decreased to 0.61 mN/m, and at a concentration of 5000 mg/L, it reduced further to 0.07 mN/m, falling within the low interfacial tension region. A solution concentration of 10,000 mg/L produced an oil–water interfacial tension of 3.32 × 10^−4^ mN/m, signifying an ultra-low interfacial tension. The significant reduction in interfacial tension enables oil–water self-emulsification with minimal disturbance. The experimental results of self-emulsification confirm that achieving oil–water spontaneous emulsification is possible by reducing interfacial tension [16,31].

### 3.4. Heavy Oil Viscosity Reduction Performance of Polymer

Figure 8a is a display of experimental results from oil–water emulsions’ viscosity and viscosity reduction rate at different polymer concentrations, considering deionized and brine water conditions. As the polymer concentration increases, the emulsion viscosity initially decreases and then increases, while the viscosity reduction rate exhibits an inverse trend. Notably, the viscosity reduction effect is more pronounced in brine water. Under brine water conditions, when the polymer concentration is 1000 mg/L, the emulsion viscosity is 7307 mPa·s, with a viscosity reduction rate of 48.96%. At this concentration, the emulsification effect is limited due to the low polymer concentration. However, as the polymer concentration rises to 3000 mg/L, the emulsion viscosity drops to 109 mPa·s, and the viscosity reduction rate reaches 99.24%. From this observation, it can be deduced that increasing the concentration enhances the emulsification effect and reduces the viscosity. At a concentration of 10,000 mg/L, the emulsion viscosity rises to 201 mPa·s with a viscosity reduction rate of 98.60%. The observed decrease and subsequent increase in emulsion viscosity and the increase and subsequent drop in the viscosity reduction rate are attributed to the increased viscosity of the water phase with higher polymer concentrations. The viscosity of O/W emulsion is determined by the water phase viscosity, where higher water phase viscosity leads to higher emulsion viscosity [39,40,41]. This phenomenon also explains why the polymer exhibits better viscosity reduction performance in brine water than in deionized water. Figure 8b displays a microscopic image of the oil–water emulsion, revealing the O/W emulsion structure formed by the polymer and heavy oil. This O/W emulsion structure significantly reduces the oil–water system’s viscosity, effectively enhancing the heavy oil’s fluidity.

Figure 9a,b depict the flow state of heavy oil without the addition of polymer. These images show that heavy oil exhibits poor fluidity. In stark contrast, Figure 9c,d show the flow state of heavy oil forming an emulsion when the polymer is added. The emulsion demonstrates exhibiting good fluidity. In Figure 9e, the viscosity–temperature profiles of both heavy oil and the heavy oil–polymer emulsion exhibit a declining trend as the temperature increases. However, an interesting observation arises with the emulsion. At 89 °C, there is a significant surge in viscosity, followed by a gradual decrease. This abrupt increase is attributed to the emulsion’s instability induced by elevated temperature, which weakens the oil–water interfacial film. Consequently, small droplets coalesce into larger ones, resulting in increased emulsion particle size and viscosity.

### 3.5. Wettability Change

Figure 10 illustrates the change in wettability of the artificial core surface before and after polymer treatment. Initially, the core surface exhibits a contact angle of 106.55°, indicating neutral wetting. However, after polymer treatment, the contact angle reduces to 34.95°, indicating a water-wet surface. The polymer treatment alters the wettability by adsorbing polymer molecules onto the core surface, with hydrophobic groups facing inward and hydrophilic groups facing outward. This transformation promotes water spreading and wetting the rock surface, facilitating the detachment of oil droplets from the core surface, and realizing oil–water replacement [19,42].

### 3.6. Polymer Oil Displacement Performance

#### 3.6.1. Sand-Packed Tube Flooding Experiments

Figure 11 presents the oil displacement curve of HPAM and PAALB, comprising three stages: water flooding, polymer flooding, and subsequent water flooding. In Figure 11a, during the water injection stage, the pressure difference initially rises to 2429 kPa before dropping rapidly to 220 kPa due to the formation of fluid advantage channels after water flooding oil breakthrough. After injecting 1.0 PV of water, the water cut increased sharply after the breakthrough, exceeding 98%, with a water flooding recovery rate of 23.22%. In the HPAM flooding stage, the pressure difference initially rises to 781 kPa and then declines, while the rate of water content rapidly drops to 55.78% before it gradually rises. The water cut surpasses 98% after injecting 1.5 PV of HPAM solution, leading to a 28.85% increase in the HPAM flooding stage recovery rate. The subsequent water flooding stage maintains a stable pressure difference and water cut, with minimal increase in the recovery rate. Finally, the recovery rate reaches 52.85% after injecting 3.0 PV of fluid.

Similarly, in Figure 11b, the water injection stage displays a rise and rapid drop in pressure difference to 2256 kPa, accompanied by a rapid increase in the water cut. After injecting 1.0 PV of water, the rate of water cut exceeds 98%, resulting in a water flooding recovery rate of 25.88%. During the PAALB flooding stage, the pressure difference initially rises to 478 kPa before declining, and the water cut drastically drops to 23.33%, followed by a gradual rise. The water cut after injection of 1.5 PV of PAALB solution was 95.44%, which increased recovery rate by 54.48%. The water cut exceeds 98% after injecting 2.0 PV of PAALB solution, leading to a 55.68% increase in the PAALB flooding recovery rate. The subsequent water flooding maintains a stable pressure difference and water cut, with a slight increase in the recovery rate. Finally, the recovery rate reaches 83.09% after injecting 3.7 PV of fluid.

#### 3.6.2. Microscopic Model Flooding Experiments

Figure 12 illustrates the microscopic mechanism of oil displacement via HPAM and PAALB. Figure 12a,d display photos and microscopic images of the model saturated with heavy oil, revealing crude oil filling the model pores. Following oil displacement via HPAM, as shown in Figure 12b,e, more residual oil remains, and fingering phenomena occur due to the significant difference in oil–water mobility. This displacement process represents the water phase pushing the oil phase forward. Figure 12c,f depict images after oil displacement by PAALB, showcasing less residual oil in the model slice and a substantial presence of fine oil droplets. This behavior suggests that PAALB disperses large oil droplets into tiny oil droplets during the displacement process, facilitating subsequent fluid displacement [43,44,45].

### 3.7. Amphiphilic Polymer Flooding Mechanism

In Figure 13, the schematic diagram showcases the mechanism of oil replacement, self-emulsification, and oil displacement by the amphiphilic polymer PAALB. Figure 13a depicts the distribution of heavy oil on the reservoir rock surface and within the rock pores. Figure 13b illustrates the oil replacement process during the initial stage of polymer injection. The polymer first adsorbs onto the rock surface, alters the wettability, and detaches oil droplets from the rock surface, dispersing the oil phase in the polymer solution. Figure 13c illustrates the process of self-emulsification during polymer oil displacement. Shear action during reservoir migration disturbs the heavy oil and polymer, leading to the self-emulsification of oil and water. The polymer molecules disperse and encapsulate large oil blocks, forming numerous tiny oil droplets and generating O/W type emulsion. Figure 13d illustrates the state after the polymer solution interacts with heavy oil. The polymer peels off and emulsifies the heavy oil from the rock surface, transforming it from initial large oil droplets to tiny oil droplets. This phenomenon improves fluidity and enhances the displacement and production of the oil phase, ultimately increasing the heavy oil recovery.

## 4. Conclusions

In summary, we synthesized an amphiphilic polymer, PAALB, through free radical polymerization, with the aim of enhancing heavy oil recovery. This versatile polymer offers multiple functionalities, including the reduction of heavy oil viscosity, oil–water displacement, and polymer flooding. A solution of 5000 mg/L PAALB effectively reduces the contact angle of water droplets on the core surface from 106.55° to 34.95°. This transition from oil-wet to water-wet conditions enables oil–water displacement. Moreover, under a 7:3 oil–water ratio, a 10,000 mg/L PAALB solution, using the glass bottle inversion method, decreases the viscosity of heavy oil from 14,315 mPa·s to 201 mPa·s, achieving an impressive viscosity reduction rate of 98.60%. In sand-packed tube flooding experiments with the same viscosity and injection volume of 1.5 PV, PAALB enhances the recovery rate by 25.63% compared to the traditional HPAM polymer.

High interfacial tension induces the contraction of oil droplets and water droplets at the oil–water interface, resulting in their separation. Notably, a 10,000 mg/L PAALB solution reduces the oil–water interfacial tension to an ultra-low level of 3.32 × 10^−4^ mN/m. This outcome mitigates the contraction tendency of droplets at the interface, promoting oil–water mixing and emulsification. It enables the self-emulsification of heavy oil and the polymer solution under weak shear conditions, transforming high-viscosity heavy oil into low-viscosity oil-in-water (O/W) emulsion, thereby improving heavy oil fluidity. Furthermore, PAALB can increase the viscosity of the water phase through hydrophobic association. This enhancement ultimately improves the flow ratio, sweep volume, and, most importantly, heavy oil recovery.

## Figures and Tables

**Figure 1 polymers-15-04606-f001:**
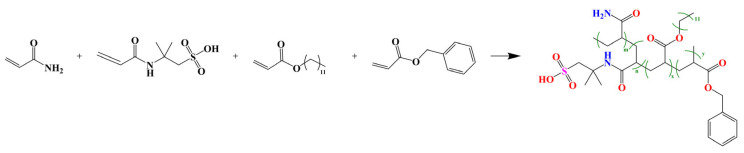
Synthesis of the amphiphilic polymer PAALB.

**Figure 2 polymers-15-04606-f002:**
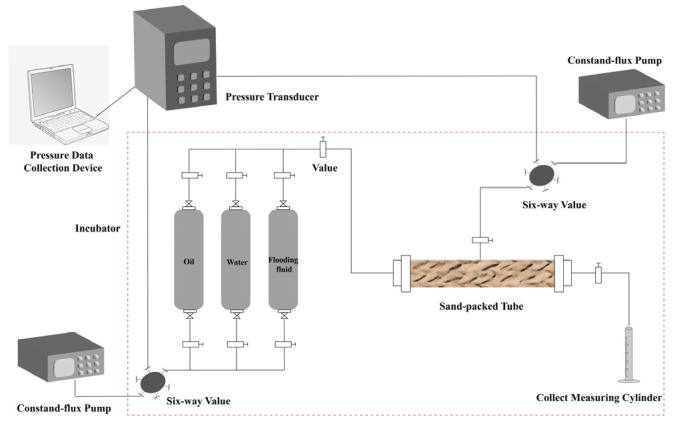
Schematic diagram of the sand-packed tube flooding experimental device.

**Figure 3 polymers-15-04606-f003:**
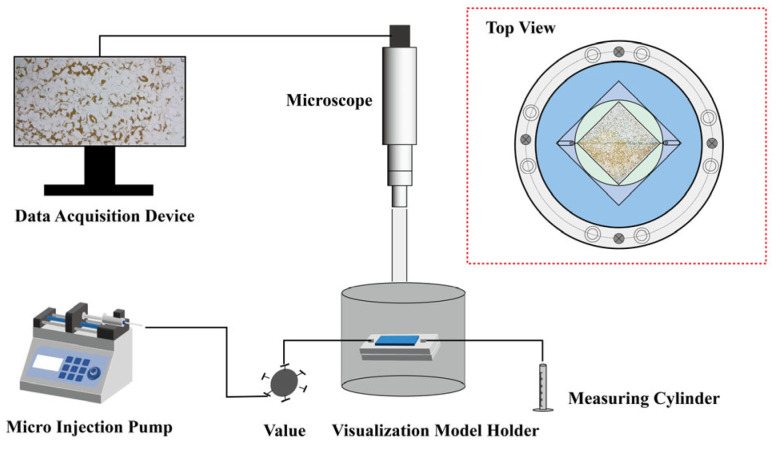
Schematic diagram of the microscopic visualization flooding experiment device.

**Figure 4 polymers-15-04606-f004:**
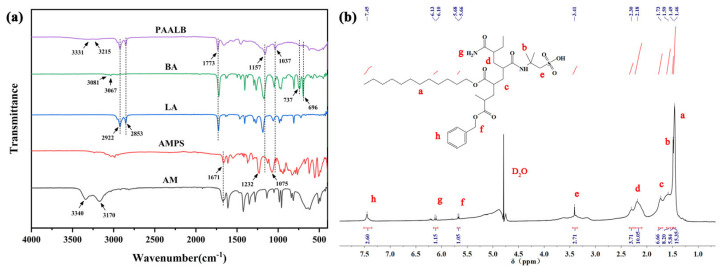
Structural characterization of PAALB (**a**) FT-IR; (**b**) ^1^H-NMR.

**Figure 5 polymers-15-04606-f005:**
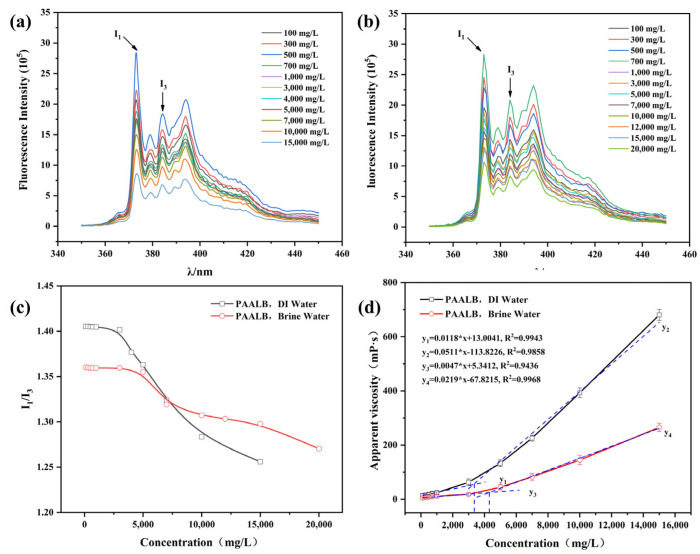
(**a**) Fluorescence emission spectrum of PAALB distilled water solution; (**b**) Fluorescence emission spectrum of PAALB brine water solution; (**c**) I_1_/I_3_ curve with PAALB Concentration; (**d**) PAALB viscosity curve with concentration.

**Figure 6 polymers-15-04606-f006:**
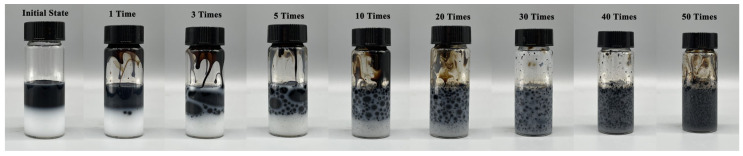
Oil–water state diagram after different inversion times.

**Figure 7 polymers-15-04606-f007:**
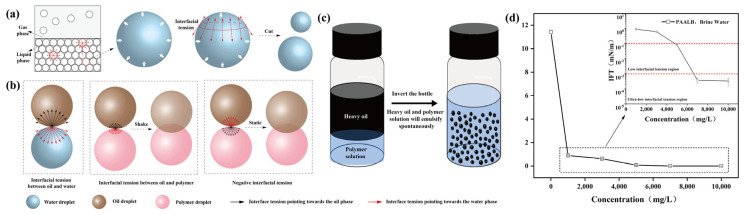
(**a**,**b**) Schematic diagram of the dynamic spontaneous emulsification mechanism; (**c**) self-emulsification of heavy oil with polymer induced by inverted bottle; (**d**) curve of interfacial tension with polymer concentration.

**Figure 8 polymers-15-04606-f008:**
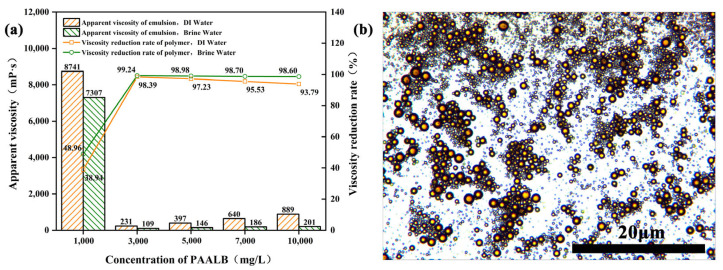
(**a**) Heavy oil viscosity reduction performance; (**b**) emulsion microscopic morphology.

**Figure 9 polymers-15-04606-f009:**
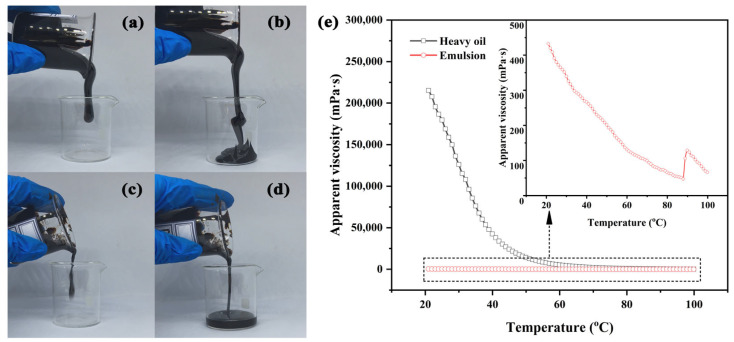
(**a**,**b**) Flowability of heavy oil; (**c**,**d**) flowability of emulsion; (**e**) viscosity–temperature curves of heavy oil and emulsion.

**Figure 10 polymers-15-04606-f010:**
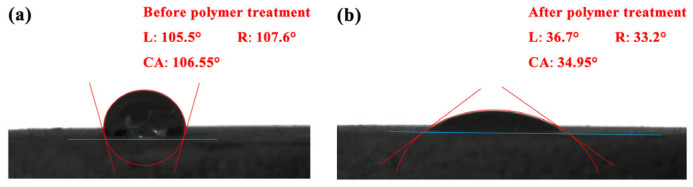
Contact angle of water with core surfaces. (**a**) before polymer treatment; (**b**) after polymer treatment.

**Figure 11 polymers-15-04606-f011:**
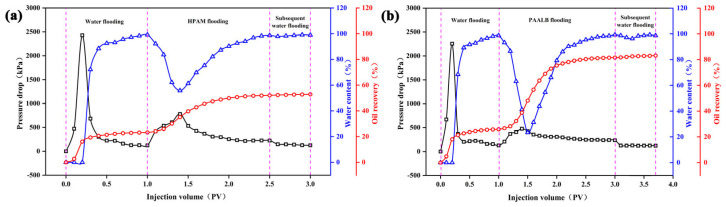
Polymer flooding curves; (**a**) HPAM; (**b**) PAALB.

**Figure 12 polymers-15-04606-f012:**
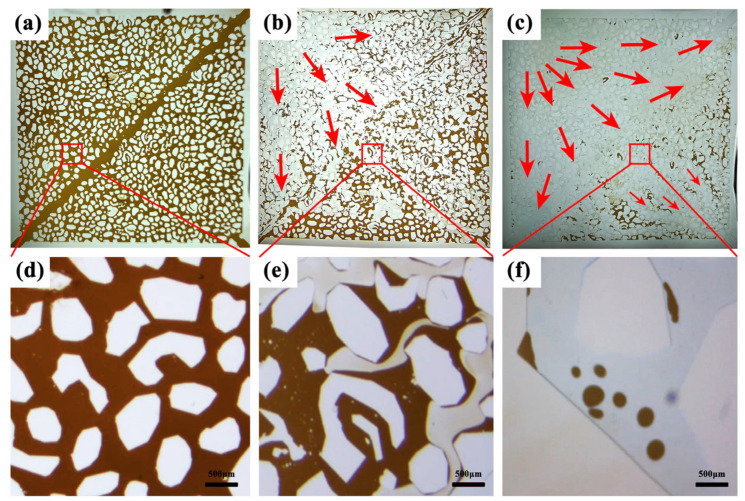
(**a**–**c**) Model image of oil saturated, HPAM flooding, and PAALB flooding; (**d**–**f**) microscopic image of oil saturated, HPAM flooding, and PAALB flooding (Scale Bars, 500 μm).

**Figure 13 polymers-15-04606-f013:**
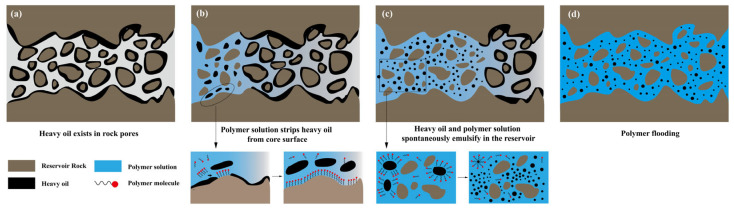
Schematic diagram of polymer mechanism for heavy oil recovery.

**Table 1 polymers-15-04606-t001:** Properties and composition of Chunfeng Heavy Oil.

Relative Density	Viscosity (50 °C)	Saturates	Aromatics	Resin	Asphaltene	Wax
0.983 g/cm^3^	14,315 mPa·s	46.17%	17.64%	10.11%	10.69%	2.62%

**Table 2 polymers-15-04606-t002:** Composition of Brine water (mg/L).

Na^+^	K^+^	Ca^2+^	Mg^2+^	Cl^−^	SO_4_^2−^	HCO_3_^−^	Total Dissolved Solids
11,500	126	6600	62	29,200	260	652	48,400

**Table 3 polymers-15-04606-t003:** Parameters of the sand-packed tube.

Polymer	Length (cm)	Diameter (cm)	Porosity (%)	Permeability (μm^2^)	Oil Saturation (%)
HPAM	30.00	2.50	36.06	5.13	89.17
PAALB	30.00	2.50	35.71	5.07	92.28

**Table 4 polymers-15-04606-t004:** Elemental analysis results of the PAALB.

Element	C	H	O	N	S
Theoretical content (wt %)	55.07	7.57	23.19	9.02	5.15
Measurement content (wt %)	50.77	7.41	28.54	8.35	5.02

**Table 5 polymers-15-04606-t005:** Gel permeation chromatography results of PAALB.

*M* _n_	*M* _w_	*M* _z_	*M* _z+1_	*M* _v_	PDI
307,883	679,306	1,048,818	1,333,611	623,887	2.21

## Data Availability

The data used to support the findings of this study are available from the corresponding author upon request.

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
