# Peer review of "Preparation and Performance Evaluation of Amphiphilic Polymers for Enhanced Heavy Oil Recovery"

_polymers, 2023, doi:10.3390/polym15234606_

Round 1

Reviewer 1 Report

Comments and Suggestions for Authors

Research is devoted to the development of methods for enhanced recovery of heavy oils. Improving the transport properties of heavy oils is also a very interesting and topical problem. Such research is important because light oil reserves are gradually being depleted. The paper is well structured and easy to read. The literature review leaves a very pleasant impression. One can go on listing its merits, but... it is not clear what it does in Polymers? The paper contains virtually no polymer content. The reviewer feels that in this case a more specialized journal such as Fuels or similar would be appropriate. If Polymers is chosen, the polymer part should be considerably expanded. In any case, the following are comments that should be considered before publication.

1. Starting on page 2, the introduction is full of polymer abbreviations that are not defined in any way. At least the monomeric base of these (co)polymers should be specified.

2. Very good lit. review, but lacks some kind of generalization, reasoning for the choice of monomers and a statement of the relationship to the purpose of the present study, which is usually placed at the end of the introduction. Instead, the authors present the results obtained at the end of the introduction, which should be placed in the abstract and/or conclusions.

3. Page. 4, first paragraph: The authors hint that there were several polymers synthesised, but test for some reason only one. If there was only one, there is no need to tease the reader with intrigue. If there were indeed several, then data for the others should be given. This would greatly improve the article and justify publication in Polymers.

4. A little further on in the same paragraph: when describing the synthesis formulation, which solutions are we talking about (solvent)?

5. Fig. 1: the copolymer formula is shown in error. a) The methylene moiety (CH2) should be included in brackets. b) Instead of lauryl acrylate (C12), pentacosane acrylate (C25) is shown.

6. For Polymers, one would like to see the initiation scheme. Also lacking is proper characterisation of the copolymer: monomer composition and molecular weight.

7. Fig. 5c: To accurately determine the CAC, there are not enough points on the horizontal parts of the curves in the low concentration region.

8. Fig. 2: "sand-packed tube" should be "sand-packed tube".

9. Figure 5, the caption contains a "stuck-on" caption for the next section. Same in figure 7 and 12.

10. Section 3.5.1 (page 12), sentence 2: seems to have lost a word.

11. Throughout the text, an error in citing figures and tables.

12. Unsuccessful choice of abbreviations for the monomers used, e.g. LAA, BAA. It is not clear what the second letter A stands for? Usually esters are labelled as LA, BA.

Best regards,

Reviewer

Reviewer 2 Report

Comments and Suggestions for Authors

In this paper, authors have synthesized a material using acrylamide, 2- acrylamido-2-methylpropane sulfonic acid, lauryl acrylate, and benzyl acrylate as starting material which has been further used in enhanced oil recovery. I have following suggestions which may be helpful to enhance the quality of manuscript. 

1. The first thing that need attention is the cost effectiveness of the synthesized materials and it should be discussed in the introduction as well in the conclusion section.

2. It is also recommended to compare the performance of the material with the previously synthesized materials on the basis of ease of synthesis and functioning of the synthesized substance.

3. Instead of figures "Error! Reference source not found." is written everywhere in the manuscript. Authors are advised to check it thoroughly.

4. In Section "2.1. Materials" the purity and product catalogue numbers of the compounds should be given. 

5. In Section 2.2., % yield of the substance is not given which is very important in synthesis case.

6. In Section 2.4, describe the methodology of viscosity measurement in detail. 

7. In most of the figures, resolution is very poor. Axes and descriptions are not visible at all. 

8. In section 2.5, was there any use of cetylpyridinium bromide which is a quencher that is used to quench the pyrene fluorescence to obtain the aggregation of amphiphilic systems?

Comments on the Quality of English Language

There are so many grammatical and typographical mistakes and authors are advised to check them thoroughly. 

Reviewer 3 Report

Comments and Suggestions for Authors

The manuscript proposes the use of a novel polymer to enhance the oil recovery.

The last part of the introduction from “in this study” to the end sounds as a summary and it does not state which is the aim and the novelty of the work. This part must be shortened and revised.

How were determined the data reported in Table 1 and Table 2?

Paragraph 2.2 How was controlled the polymerization of the polymer? How the different monomers can react to give the reported polymer architecture? The rationale about the synthesis must be better explained. The information on the molecular weight of the polymer must be provided as well as the monomer ratio. The amphiphilicity of the polymer must be better explained.

Paragraph 2.4 The method for the measurement of viscosity must be better reported. Which concentrations were tested? Why 7.34 s-1? Was viscosity measured only at one shear rate values or in a range of shear rates and stresses?

Paragraph 2.6 How many times the bottle was inverted?

Paragraph 2.8 Which magnification was used for microscope images?

Paragraph 2.9 Which concentrations were tested? More information on the method should be provided.

Paragraph 2.10 What core slices are? How the contact angle was measured?

Paragraph 3.1 The FT-IR spectrum of the monomer and physical mixture of the monomers should be provided to draw any conclusions about the peak shift. Integration of the NMR signals should be provided to assess the formation of the polymer.

Paragraph 3.2 In the method section should be provided how the CAC concentrations were determined and which method used for the fitting of the data.

It is unusual that the CAC of the polymers in brine is higher than in water. The authors provided a possible explanation but possibly some references to support this should be added, since generally salts have a different effect on the amphiphiles behaviour.

Paragraph 3.2.2 The measured interfacial values are very low between water and the oil without the polymers and very very low with the polymer. Can be they reliably measured using the reported instrument (spinning drop tensiometer)?

Round 2

Reviewer 1 Report

Comments and Suggestions for Authors

The authors wrote a couple more papers answering the reviewers' questions. The reviewer thanks the authors for the extremely detailed answers. Absolutely all issues have been resolved. Only Figure 1 needs to be corrected (see attached file).

Reviewer 2 Report

Comments and Suggestions for Authors

The paper is now acceptable in Polymers

Author Response

We greatly appreciate your contribution to the review of this manuscript.

Reviewer 3 Report

Comments and Suggestions for Authors

The manuscript is suitable for publication

Author Response

(The authors gave the same response as above.)
